# Efficacy of HPV Vaccination in Women Receiving LEEP for Cervical Dysplasia: A Single Institution’s Experience

**DOI:** 10.3390/vaccines8010045

**Published:** 2020-01-25

**Authors:** Marco Petrillo, Margherita Dessole, Elettra Tinacci, Laura Saderi, Narcisa Muresu, Giampiero Capobianco, Antonio Cossu, Salvatore Dessole, Giovanni Sotgiu, Andrea Piana

**Affiliations:** 1Biomedical Sciences, University of Sassari, 07100 Sassari, Italy; narcisamuresu@outlook.com (N.M.); capobia@uniss.it (G.C.); dessole@uniss.it (S.D.); gsotgiu@uniss.it (G.S.); piana@uniss.it (A.P.); 2Gynecologic and Obstetric Clinic, University of Cagliari, 09126 Cagliari, Italy; margheritadessole@gmail.com; 3Gynecologic and Obstetric Clinic, Department of Clinical and Experimental Medicine, University of Sassari, 07100 Sassari, Italy; elettra.tinacci@live.com; 4Clinical Epidemiology and Medical Statistics Unit, Deptartment of Medical, Surgical and Experimental Sciences, University of Sassari, 07100 Sassari, Italy; lsaderi@uniss.it; 5Surgical Pathology, Department of Medical, Surgical and Experimental Sciences, University of Sassari, 07100 Sassari, Italy; cossu@uniss.it; 6Section of Hygiene and Preventive Medicine, Department of Medical, Surgical and Experimental Sciences, University of Sassari, 07100 Sassari, Italy

**Keywords:** HPV vaccine, LEEP, cervical dysplasia, recurrence, prevention

## Abstract

The aim of this study was to assess the role of a human papilloma virus (HPV) vaccine after loop electrosurgical excision procedure (LEEP) in reducing recurrent cervical dysplasia. A series of 503 women with cervical dysplasia received LEEP between January 2012 and October 2018. Of these patients, 379 were treated between January 2012 and June 2017, thus ensuring an adequate follow-up time. We made three attempts to establish telephone contact with each patient; 77 women did not respond and were excluded from the final study population, which consisted of 302 patients. One hundred eighty-two (60.7%) women were vaccinated with an HPV vaccine within 4 weeks of LEEP and 103 (34.3%) were followed up with but not vaccinated. Recurrence of cervical dysplasia requiring a further LEEP procedure occurred in 30 (10.5%) women, of whom 17 (16.5%) were not vaccinated and 13 (7.1%) were vaccinated (*p*-value = 0.010). At univariate analysis, HPV vaccination after LEEP (odds ratio (OR) = 0.4, *p*-value = 0.020) emerged as an independent protective factor. Choosing as an outcome of the analysis only recurrence as severe cervical lesions, the protective role of HPV vaccination after LEEP was found to be much more relevant with an odds ratio of 0.2 (95% CI = 0.1–0.6, *p*-value = 0.02). Administration of an HPV vaccine after LEEP seems to reduce the risk of recurrence, thus suggesting that HPV vaccination has a role as an adjuvant treatment after LEEP.

## 1. Introduction

Human papilloma virus (HPV) causes one of the most prevalent sexually transmitted infections, and it has been estimated that around 80% of women will be exposed to HPV during their lifetime [1]. Despite the high circulation of HPV genotypes, the majority of women develop an asymptomatic infection, without any clinical or histological signs [2]. However, persistent HPV infection may cause noninvasive and invasive cancer [2].

At the beginning of the 20th century, cervical cancer represented the second most common incident cause of cancer-related deaths in the female population in the United States (40 deaths per 100,000 females). The mortality rate decreased to 2.3 deaths per 100,000 females in 2019 [3], following the implementation and scale-up of primary and secondary prevention strategies (i.e., HPV vaccine and Papanicolaou smear testing, respectively) to early detect precancerous cervical lesions [2,3,4,5,6,7,8,9].

If the introduction of pap smears represents the first milestone in the fight against HPV-related malignancies, the second milestone is certainly represented by the development of HPV vaccines [2,3,4,5,6,7,8,9]. In particular, from 2006 to 2014, three vaccines containing virus-like particles (VLPs) were licensed as effective primary prevention strategies for cervical cancer and genital warts [10,11], and it has been estimated that a 70% reduction in disease burden will occur over the next four decades in high-income countries if primary prevention strategies are applied accurately [12]. On the other hand, the therapeutic role of HPV vaccines remains highly debated. It has been hypothesized that the vaccine-related induction of an immunological response against HPV may favour virus clearance; however, data regarding this specific point remain inconclusive. In particular, initial studies reported that HPV vaccine administration after loop electrosurgical excision procedure (LEEP) of cervical dysplasia may reduce the recurrence rate [13]. However, a further randomized clinical trial did not confirm the previous findings, suggesting that HPV vaccine administration after LEEP in women with cervical dysplasia confers no benefit [14]. Taken together, these results emphasize the lack of a definitive conclusion on this specific point.

Therefore, with the aim of clarifying whether HPV vaccination after LEEP may be associated with a reduced recurrence rate in women with cervical dysplasia, we conducted this single-institution observational retrospective study.

## 2. Materials and Methods

### 2.1. Patients

We analyzed a series of women with a confirmed diagnosis of cervical dysplasia who underwent LEEP between January 2012 and October 2018 at the Obstetrics and Gynecology Unit of an Italian university hospital. Only women with ≥2 years of follow-up were enrolled on the assumption that ~90% of post-LEEP recurrences occur within the first 24 months after surgery [15,16,17,18] (Figure 1). Colposcopy with a biopsy was performed to confirm abnormal cytological findings; however, HPV testing, with virotype characterization, was not routinely performed in the time period 2012–2016; therefore, in several cases, the results of HPV testing were reported only as the presence of high-risk genotype. The surgical procedure was performed using colposcopic visualization of cervical lesions under local anesthesia at an outpatient clinic. Abnormal cervical tissue was removed using a wire loop that was heated by an electric engine.

After January 2015, all women were counseled with regard to the potential benefits and risks of HPV vaccination and, after acceptance, received within 4 weeks after LEEP the first dose of an HPV vaccine, which was administered at the local general community hospital. Only the first three women who enrolled in the study received the bivalent vaccine; all other patients that were included in the study were administered the quadrivalent HPV vaccine.

The clinical follow-up included a pap smear and a colposcopy every 6 months for the first 2 years after LEEP. Furthermore, in accordance with international guidelines, if margins were positive, then cytology and endometrial curettage at the 3-month evaluation were performed.

All women who participated in the study were contacted by telephone and interviewed on oral contraceptive prescriptions, smoking habits, total number of sexual partners, number of pregnancies, concomitant drugs, and occurrence of infectious diseases.

A treatment outcome failure was defined as a patient with ≥1 recurrence of cervical dysplasia after LEEP.

HPV genotype at the time of relapse was assessed only in the group of vaccinated women on histological specimens obtained at the time of recurrence.

According to Italian law on observational studies, ethical approval is not needed for the analysis of anonymous retrospective data. Institutional Review Board (IRB) approval was obtained (IRB code 2018-375).

### 2.2. LEEP Procedure and Histological Findings

LEEP was performed under local anesthesia using wire loop electrodes with a diathermy apparatus set. A suture was placed at the 12 o’clock position of the specimen for orientation before fixation in 10% formalin. Cervical dysplasia grading and a histologic assessment of exocervical and endocervical margins of resection were carried out; in the case of involvement of surgical margins, clinical monitoring was performed, followed by surgical excision for persistent severe dysplasia. Furthermore, cone length and p16 expression were assessed in all specimens.

### 2.3. Statistical Analysis

An ad hoc electronic form was prepared to collect qualitative and quantitative variables from medical charts. Qualitative variables are described with absolute and relative (percentage) frequencies, whereas quantitative variables are summarized with means (standard deviations, SD) or medians (interquartile ranges, IQRs), depending on their parametric distribution. Comparisons between vaccinated and nonvaccinated patients were performed with a chi-squared or Fisher’s exact test for qualitative variables, whereas Student’s *t* and Mann–Whitney tests were performed for parametric and nonparametric variables, respectively.

A logistic regression analysis was carried out to assess the relationship between recurrence rate and demographic, epidemiological, and clinical data.

A two-tailed *p*-value less than 0.05 was considered to be statistically significant. All statistical analyses were performed with the statistical software STATA version 15 (StataCorp, Texas, TX, USA).

## 3. Results

### 3.1. Study Sample

A total of 503 women underwent LEEP due to cervical dysplasia during the study period. However, only 379 patients were followed-up with for ≥2 years. Seventy-seven (20.3%) women did not respond to our telephone calls and were excluded from the analysis (Figure 1).

A total of 182 (63.8%) women received an HPV vaccination within 4 weeks after LEEP. The remaining 103 (36.2%) patients entered a routine follow-up program but did not receive an HPV vaccine (Figure 1).

As described in Table 1, the median (IQR) age was 39 (32–47) years old, with ~24% of patients reporting smoking habits (Table 1). The vast majority (72.3%) of women were of middle–upper socio-economic status, with >3 sexual partners over their lifetime (68.4%). Almost all women underwent LEEP for moderate/severe cervical intraepithelial neoplasia (CIN2/CIN3/carcinoma in situ: 95.1%). Involvement of margin of resection occurred in 25 (8.8%) patients, and a positive cone apex was found in 43 (15.1%) women, which appears to be superimposable with data from the available literature [18]. Recurrent cervical dysplasia occurred in 30 (10.5%) patients, with CIN2/CIN3/carcinoma in situ in 20 (66.7%) women, with no clinically significant differences in terms of follow-up between the two groups. As described above, in the majority of women (83.4%, Table 1) HPV testing reported just the presence of high-risk HPV genotypes, without further characterization. However, we did not observe differences between vaccinated and nonvaccinated patients in the distribution of HPV virotypes (Table 1).

### 3.2. Effectiveness of Post-LEEP HPV Vaccination

Women administered an HPV vaccine after LEEP were 4 years younger when compared with nonvaccinated patients (41 vs. 37.5 years; *p*-value = 0.0004; Table 1).

Disease recurrence occurred in 17 (16.5%) and 13 (7.1%) nonvaccinated and vaccinated women, respectively (*p*-value = 0.010) (Table 2). A statistically significant lower incidence of CIN2/CIN3/carcinoma in situ recurrence was found in the vaccinated group (13.6% vs. 3.3%; *p*-value = 0.001).

HPV genotype at the time of relapse was retrieved in five women receiving vaccination after LEEP. In particular, two patients showed HPV 16 infection (both with HPV 16 at diagnosis), one woman was affected by HPV 33 (HPV 33,59 at diagnosis), in one patient we detected HPV 51 (high-risk HPV at diagnosis), and in the remaining woman we observed HPV 31 infection (high-risk HPV at diagnosis).

### 3.3. Predictors of LEEP Failure

Table 3 shows that only patients vaccinated after LEEP had a decreased risk of recurrences (odds ratio (OR) = 0.4; *p*-value = 0.020). The protective role of HPV vaccination after LEEP was found to be more relevant to protection against severe (CIN3/carcinoma in situ) cervical lesions (OR = 0.2; *p*-value = 0.02).

## 4. Discussion

The successful implementation of HPV vaccination strategies has significantly reduced the incidence of cervical cancer [12]. However, several authors have disputed the potential role of HPV vaccination as an adjuvant treatment after surgical excision [19,20,21]. The results of our study demonstrate, in a real-life scenario, that the introduction of post-LEEP vaccination is associated with a reduction in recurrence rate.

In particular, contrasting results on the risk of recurrence have been described in women treated with LEEP for a cervical dysplasia [13,14]. In 2013, Kang et al. showed a statistically significantly lower recurrence rate in a large sample of >700 women administered the HPV quadrivalent vaccine after LEEP [13]. Furthermore, the recently published randomized clinical trial also known as SPERANZA study demonstrated a clinical effectiveness of 80% in disease relapse prevention after HPV vaccine administration in LEEP-treated women following a diagnosis of CIN2 [22]. Similar findings were described by the same Italian group, who showed a reduced relapse rate (from 13% to 3.5%) in 178 vaccinated and HPV-negative surgically treated women [23].

On the other hand, in 2016, a randomized clinical trial that involved 1711 women, each of whom either received or did not receive an HPV vaccine after surgical excision, did not confirm previously reported data. The authors concluded that vaccination does not protect against infections/lesions after treatment [14]. Therefore, as pointed out by several authors, the role of vaccination as an adjuvant treatment after surgery for HPV-related cervical lesions remains questionable [2].

In this context, the results of our study should not be underestimated. In fact, we demonstrated in a series of more than 300 women a statistically significant reduction of recurrence rate in women receiving a quadrivalent HPV vaccination after LEEP when compared to women who did not receive a vaccine, thus confirming previous results [14]. Furthermore, as reported in Table 4, the benefit of HPV vaccination seems to be particularly relevant to a reduction in the occurrence of severe cervical lesions.

Therefore, while our results do not enable us to draw definitive conclusions, they do support the introduction of post-LEEP HPV vaccination in women receiving an excisional procedure due to cervical dysplasia, particularly in the case of severe lesions. It seems reasonable to hypothesize that, particularly considering that the results from randomized clinical trials did not demonstrate an increase in virus clearance after HPV vaccine administration [14], the benefit observed in our study and the abovementioned studies could be related to the prevention of reinfection in women with complete HPV clearance after LEEP. However, we cannot exclude the possibility that the benefit of post-LEEP vaccination is not only related to the prevention of reinfection, particularly considering the evidence in favor of newly developed therapeutic vaccines [24,25]. Future studies are required to answer this crucial question.

It should be acknowledged that, in our study, the cohort of patients who received an HPV vaccine showed a statistically significantly lower age as compared to women who did not receive an HPV vaccine, thus potentially influencing the rate of recurrence that was observed in the two groups. However, the results of the univariate analysis, from which HPV quadrivalent vaccine administration after LEEP emerged as the only predictor of failure after LEEP (Table 4), strongly confirm the reliability of our data.

The small sample size in comparison with other international studies and the high drop-out rate in our cohort could limit the reliability of our findings. Furthermore, the lack of HPV testing with genotype characterization before and after LEEP meant that we could not clarify the role that HPV plays in HPV-positive patients after LEEP intervention. However, the scientific evidence from our observational study, which is characterized by homogeneous groups for the main confounding variables, could help us to evaluate the preventive/therapeutic role of HPV vaccines in surgically treated patients.

## 5. Conclusions

In conclusion, the administration of an HPV vaccine after LEEP reduces the risk of recurrence, suggesting that HPV vaccines can effectively be used as an adjuvant treatment after an excisional procedure. Further controlled clinical trials are needed to better elucidate the efficacy of the currently available HPV vaccines in different epidemiological scenarios.

## Figures and Tables

**Figure 1 vaccines-08-00045-f001:**
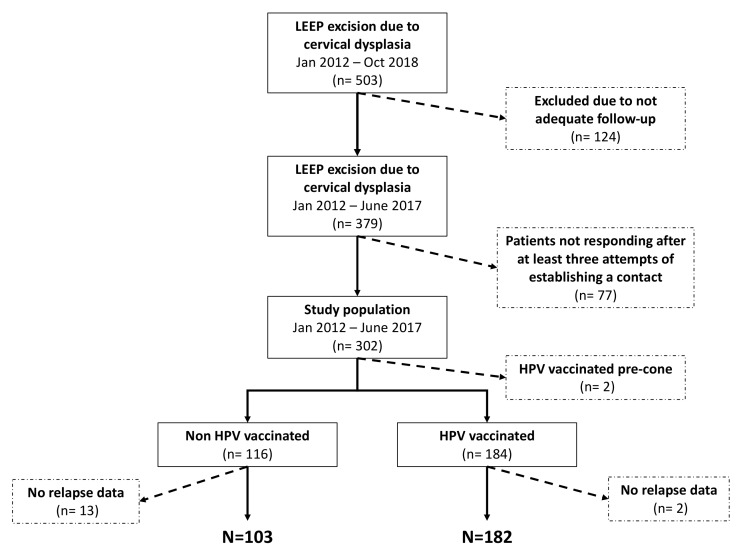
A consort diagram of our study population.

**Table 1 vaccines-08-00045-t001:** Comparison of epidemiological and clinical variables among vaccinated and nonvaccinated women.

Clinico-Pathological Variables		Total	Nonvaccinated	Vaccinated	*p*-Value
All cases		285 (100.0)	103 (36.1)	182 (63.9)	-
Median (IQR) age, years		39 (32–47)	41 (36–49)	37.5 (30–44)	0.0004
Oral contraceptive use, *n* (%)		46 (16.1)	16 (13.8)	32 (17.4)	0.41
Infections, *n* (%)		6 (2.1)	3 (2.6)	3 (1.6)	0.68
Vegetarian habits, *n* (%)		16 (5.6)	7 (6.0)	11 (6.0)	1.0
Smoking exposure, *n* (%)		69 (24.2)	28 (24.1)	46 (25.0)	0.87
Total number of sexual partners >3, *n* (%)		195 (68.4)	77 (66.4)	129 (70.1)	0.50
Cervical biopsy, *n* (%)	Negative	2 (0.7)	1 (0.9)	1 (0.6)	0.98
	CIN1	5 (1.8)	2 (1.8)	3 (1.7)	
	CIN2	144 (51.8)	57 (50.9)	96 (53.0)	
	CIN3	113 (40.7)	45 (40.2)	72 (39.8)	
	CIS	14 (5.0)	7 (6.3)	9 (5.0)	
HPV virotypes, *n* (%)	HPV 16	17 (5.9)	4 (3.9)	13 (7.1)	0.484
	HPV 16 + other	8 (2.9)	1 (0.2)	7 (3.8)	
	HPV 18	2 (0.3)	0 (0.0)	2 (1.1)	
	HPV 31	9 (3.3)	3 (2.9)	6 (3.3)	
	HPV 51	4 (1.6)	2 (1.9)	2 (0.1)	
	Other	8 (2.6)	4 (3.9)	4 (12.8)	
	High-risk *	237(83.4)	89 (87.2)	148 (71.8)	
Immunodepression, *n* (%)		1 (0.4)	-	1 (0.5)	1.0
Median (IQR) cone length, mm		25 (15–35)	25 (15–35)	25 (15–30)	0.62
Histology cone, *n* (%)	Negative	9 (3.2)	2 (1.7)	8 (4.4)	0.44
	CIN1	43 (15.1)	14 (12.1)	31 (16.9)	
	CIN2	104 (36.5)	45 (38.8)	65 (35.3)	
	CIN3	103 (36.1)	41 (35.4)	65 (35.3)	
	CIS	26 (9.1)	14 (12.1)	15 (8.2)	
P16 positive, *n* (%)		91 (31.9)	37 (31.9)	58 (31.5)	0.95
Involvement of glandular outlets, *n* (%)		129 (45.3)	58 (50.0)	76 (41.3)	0.14
Positive margin of resection, *n* (%)		25 (8.8)	15 (12.9)	13 (7.1)	0.09
Positive apex, *n* (%)		43 (15.1)	23 (19.8)	26 (14.1)	0.19
Median follow-up time, months (range)		6 (4–12)	6 (5–12)	6 (3–8)	0.002

HPV: human papilloma virus. *: Further HPV genotype characterization not available.

**Table 2 vaccines-08-00045-t002:** Comparison of recurrence rate and histological findings between vaccinated and nonvaccinated patients.

Clinico-Pathological Variables		Nonvaccinated *n* = 103	Vaccinated *n* = 182	*p*-Value
Relapse		17 (16.5)	13 (7.1)	0.01
Histology at recurrence, *n* (%)	CIN1	3/17 (17.7)	7/13 (53.9)	0.04
	CIN2	6/17 (35.3)	4/13 (30.8)	0.80
	CIN3	8/17 (47.1)	1/13 (7.7)	0.02
	CIS	0/17 (0.0)	1/13 (7.7)	0.25

**Table 3 vaccines-08-00045-t003:** Logistic regression analysis to assess the relationship between relapse (any histologic type) and epidemiological and clinical variables.

Clinico-Pathological Variables	Relapse	
	OR (95%CI)	*p*-Value
Age ≥40	1.2 (0.6–2.6)	0.60
Age	1.0 (1.0–1.1)	0.33
Positive margin of resection	2.3 (0.8–6.5)	0.13
Positive apex	1.4 (0.6–3.8)	0.46
Oral contraceptive use	1.1 (0.4–2.9)	0.92
Smoking exposure	1.1 (0.5–2.7)	0.76
Parity	1.1 (0.8–1.6)	0.59
Vaccinated	0.4 (0.2–0.8)	0.02

**Table 4 vaccines-08-00045-t004:** Univariate analysis.

Clinico-Pathological Variables	Relapse	
	OR (95%CI)	*p*-Value
Age ≥40	1.2 (0.6–2.6)	0.60
Age	1.0 (1.0–1.1)	0.33
Positivity margin of resection	2.3 (0.8–6.5)	0.13
Apex positivity	1.4 (0.6–3.8)	0.46
Oral contraceptive use	1.1 (0.4–2.9)	0.92
Smoking exposure	1.1 (0.5–2.7)	0.76
Parity	1.1 (0.8–1.6)	0.59
Vaccinated	0.4 (0.2–0.8)	0.02

For outcome severe cervical lesion (CIN3/Carcinoma in situ); OR (95%CI): 0.2 (0.1–0.6); *p*-value = 0.02.

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
