# Peer review of "Efficacy of HPV Vaccination in Women Receiving LEEP for Cervical Dysplasia: A Single Institution’s Experience"

_vaccines, 2020, doi:10.3390/vaccines8010045_

Round 1
Reviewer 1 Report
Petrillo M et al demonstrated the efficacy of HPV vaccination after LEEP procedure in the retrospective analysis. As the authors mentioned in the discussion, there are many previous studies regarding this issue. One of the main lack of study is that there is no information about HPV infection in the study population. Also, the positivity rate of p16 is around 30%. p16 status has been considered to discriminate HPV positiveness.
Please provide the information on HPV infection status (genotypes, pre and post LEEP) in the study population. Please add the information about what kind of vaccines they injected after LEEP. How many times? and when did they receive it?
Author Response
Comment 1. Petrillo M et al demonstrated the efficacy of HPV vaccination after LEEP procedure in the retrospective analysis. As the authors mentioned in the discussion, there are many previous studies regarding this issue. One of the main lack of study is that there is no information about HPV infection in the study population. Also, the positivity rate of p16 is around 30%. p16 status has been considered to discriminate HPV positiveness. Please provide the information on HPV infection status (genotypes, pre and post LEEP) in the study population.
Answer: We thank the Reviewer#1 for his/her useful comments and for having provided important points which could help us improve the quality of the present manuscript. We would have liked to provide information on the genotypes. Unfortunately, we could only discuss about HPV-positivity/negativity. We did not collect this important information and we cannot retrospectively retrieve it. We included this key limitation in the Discussion section: this could hinder the reliability of the estimation of the relapse rate and of the new infection/s by different genotypes during the follow-up.
Comment 2. Please add the information about what kind of vaccines they injected after LEEP. How many times? and when did they receive it?
Answer: We thank the Reviewer for his/her request of clarification on the immunization status. The first dose was prescribed and administered within the first month after LEEP. The schedule of the vaccine should have been that recommended by the international and national regulatory agencies (i.e., three doses). The vast majority of women has reported in the revised version of the paper (Lines 82-83).

Reviewer 2 Report
This was a retrospective cohort study that evaluated the efficacy of HPV vaccine for reducing recurrent cervical dysplasia among women who underwent loop electrosurgical excision procedure (LEEP) for cervical dysplasia. Women with at least 2-years of follow-up were included. The authors found a 60% reduction in cervical dysplasia recurrence among women who received the HPV vaccine after LEEP compared to those who did not, and an 80% reduction in severe cervical lesions. This was a nice study and the findings are an important contribution to the prevention of recurrent cervical dysplasia. Below are my comments.
The manuscript needs to undergo extensive editing for English language, grammar, and sentence structure.
Title: HPV and LEEP should be capitalized in the title.
HPV is never defined. Check the journal guidelines whether this is an accepted commonly used abbreviation.
Line 24 and Line 110: the abstract and results states that 182 women received the quadrivalent HPV vaccine but table 1 states that the n=181. Please resolve this discrepancy.
Lines 28-29: The odds ratio shown is for receipt of HPV vaccine, not lack of HPV vaccine. As stated, not receiving vaccine is protective.
Lines 30-31: It is not clear what the referent category is for this analysis. No recurrence or no recurrence plus CIN1/CIN2. I will discuss this point further later in this review.
Line 57: LEEP is not defined properly in the manuscript.
Lines 71-72: Please include which HPV vaccine the women received in the methods.
Line 102: the reference for Stata is “(StataCorp, Texas, USA)” not “StatsCorp”.
Results: It appears that not all women had a positive result for cervical dysplasia at time of LEEP (Tables 1 and 2). Please clarify this observation and explain why they were not excluded from the analyses.
Table 1 repeats the information that is contained in Table 2. Table 2 is more informative and appropriate for this data so Table 1 should be deleted.
Line 116: Clarify whether number of sexual partners were lifetime, past year, etc.
Please add the number of women for each group in the column headings for Table 2.
Please add the median time of follow-up for each group to the results section text as this is an important point since HPV vaccination was not recommended until 2015.
Lines 129, 130, 138: It is not clear what the referent category is for these analyses. No recurrence or no recurrence plus CIN1/CIN2 (or no recurrence plus CIN1). The authors are potentially diluting the effect of vaccine if women with less sever dysplasia are lumped in with no dysplasia so severe dysplasia should be compared to no dysplasia. I am having trouble recreating these results so the authors need to clarify these analyses.
Table 4. Table 2 also represent unadjusted analyses and many fo the factors shown were already shown to not be associated with recurrence so it is not necessary to include them. Age was significant in Table 2 so I question the non-significant result in Table 4 unless the authors are showing adjusted results instead of unadjusted (univariate) results. The authors need to find a better way of displaying and explaining the results in Tables 3 & 4 to reflect the analyses that were done.
Author Response
Comment 1. This was a retrospective cohort study that evaluated the efficacy of HPV vaccine for reducing recurrent cervical dysplasia among women who underwent loop electrosurgical excision procedure (LEEP) for cervical dysplasia. Women with at least 2-years of follow-up were included. The authors found a 60% reduction in cervical dysplasia recurrence among women who received the HPV vaccine after LEEP compared to those who did not, and an 80% reduction in severe cervical lesions. This was a nice study and the findings are an important contribution to the prevention of recurrent cervical dysplasia. Below are my comments.
Answer. We thank the Reviewer #2 for his/her fruitful and helpful comments. The suggestions and recommendations of both reviewers improved the scientific quality of the manuscript.
Comment 2. The manuscript needs to undergo extensive editing for English language, grammar, and sentence structure.
Answer. We thank the Reviewer#2 for this suggestion. The English language was carefully revised using the publisher official service.
Comment 3. Title: HPV and LEEP should be capitalized in the title.
Answer. We thank for the suggestion: both words were capitalized.
Comment 4. HPV is never defined. Check the journal guidelines whether this is an accepted commonly used abbreviation.
Answer. We thank for the suggestion; acronyms were spelled out to avoid any misunderstandings.
Comment 5. Line 24 and Line 110: the abstract and results states that 182 women received the quadrivalent HPV vaccine but table 1 states that the n=181. Please resolve this discrepancy.
Answer. We thank the Reviewer, and the sections of the manuscript were carefully revised.
Comment 6. Lines 28-29: The odds ratio shown is for receipt of HPV vaccine, not lack of HPV vaccine. As stated, not receiving vaccine is protective.
Answer. We thank the Reviewer for having raised this issue; the abstract section was edited accordingly, clarifying the protective role of the HPV vaccination (the OR is less than one and the outcome variable is the occurrence of a relapse).
Comment 7. Lines 30-31: It is not clear what the referent category is for this analysis. No recurrence or no recurrence plus CIN1/CIN2. I will discuss this point further later in this review.
Answer. We thank the Reviewer. The variable we considered in the analyses was the occurrence of relapse (any histological grade).
Comment 8. Line 57: LEEP is not defined properly in the manuscript.
Answer. We thank for the request of clarification. We revised the Methods section including more information on the surgical procedure we adopted.
Comment 9. Lines 71-72: Please include which HPV vaccine the women received in the methods.
Answer. We thank the Reviewer for this request of clarification. We detailed the type of vaccine prescribed to the cohort of women we followed-up.
Comment 10. Line 102: the reference for Stata is “(StataCorp, Texas, USA)” not “StatsCorp”.
Answer. We thank the Reviewer for the suggestion; we edited the typo, accordingly.
Comment 11. Results: It appears that not all women had a positive result for cervical dysplasia at time of LEEP (Tables 1 and 2). Please clarify this observation and explain why they were not excluded from the analyses.
Answer. We thank the Reviewer for having raised this important issue. We edited the figures included in the current Table 1 to display only findings related to vaccinated and non-vaccinated women.
Comment 12. Table 1 repeats the information that is contained in Table 2. Table 2 is more informative and appropriate for this data so Table 1 should be deleted.
Answer. We really thank the Reviewer for this suggestion. Table no. 1 was deleted.
Comment 13. Line 116: Clarify whether number of sexual partners were lifetime, past year, etc.
Answer. We thank the Reviewer for this request of clarification. We detailed that the above-mentioned variable refers to the number of lifetime partners.
Comment 14. Please add the number of women for each group in the column headings for Table 2.
Answer. We thank the Reviewer. The new table no. 1 was edited and the denominators were included in the first row.
Comment 15. Please add the median time of follow-up for each group to the results section text as this is an important point since HPV vaccination was not recommended until 2015.
Answer. We thank the Reviewer for this request of clarification. The information is now displayed in the Table no. 1.
Comment 16. Lines 129, 130, 138: It is not clear what the referent category is for these analyses. No recurrence or no recurrence plus CIN1/CIN2 (or no recurrence plus CIN1). The authors are potentially diluting the effect of vaccine if women with less sever dysplasia are lumped in with no dysplasia so severe dysplasia should be compared to no dysplasia. I am having trouble recreating these results so the authors need to clarify these analyses.
Answer. We thank the Reviewer for having raised this important issue. As we detailed in a previous response, we did not combine the occurrence of mild cervical lesions with no recurrence. All the analyses were focused on the occurrence of relapse (any grade) VS. the occurrence of no relapse.
Comment 17. Table 4. Table 2 also represent unadjusted analyses and many fo the factors shown were already shown to not be associated with recurrence so it is not necessary to include them. Age was significant in Table 2 so I question the non-significant result in Table 4 unless the authors are showing adjusted results instead of unadjusted (univariate) results. The authors need to find a better way of displaying and explaining the results in Tables 3 & 4 to reflect the analyses that were done.
Answer. We thank the Reviewer for this comment. We edited the title of the univariate analysis to better clarify the aim of the analysis: we wanted to assess the relationship between the outcome variable (i.e., occurrence of any cervical relapse) and independent epidemiological and clinical variables. The statistically significant difference for the variable age in the current Table no. 1 can be attributed only to the difference of the median age in the vaccinated and non-vaccinated group; then, this statistical result does not have any role in the analysis of the logistic regression described in the Table no. 3.

Round 2
Reviewer 1 Report
I regret to say it is not an informative study without the information about their HPV infection status. Although the types of HPV vaccines have been demonstrated in the text, we still don't know the real effect of vaccination without HPV status. The manuscript is well written but there is a lot of similar articles in the world. It is not acceptable in Vaccines at it stands. I look forward to seeing the next study with the status of HPV genotypes of the population.
Author Response
Reviewer#1:
Comment 1. I regret to say it is not an informative study without the information about their HPV infection status. Although the types of HPV vaccines have been demonstrated in the text, we still don't know the real effect of vaccination without HPV status. The manuscript is well written but there is a lot of similar articles in the world. It is not acceptable in Vaccines at it stands. I look forward to seeing the next study with the status of HPV genotypes of the population.
Answer: We thank the Reviewer#1 for his/her useful comments. As requested, data regarding HPV infection status have been reported in Table 1, and throughout the manuscript (lines 134-136; lines 149-153).
Round 3
Reviewer 1 Report
I have no concern about the manuscript. All the points are improved.